# A Differentiable self-disambiguated sense embedding model via Scaled Gumbel Softmax

## Abstract

We present a differentiable multi-prototype word representation model that disentangles senses of polysemous words and produces meaningful sense-specific embeddings without external resources. It jointly learns how to *disambiguate* senses given local context and how to *represent* senses using hard attention. Unlike previous multi-prototype models, our model approximates differentiable discrete sense selection via a modified Gumbel softmax. We also propose a novel human evaluation task that quantitatively measures (1) how meaningful the learned sense groups are to humans and (2) how well the model is able to disambiguate senses given a context sentence—an evaluation ignored by previous models. Our model not only discovers distinct, interpretable embeddings but is competitive against previous models on word similarity tasks.

## 1 Sense-specific embedding

Machine learning models for natural language processing applications often represent words with real valued vector *embeddings*. Popular word embedding models such as Word2Vec (Mikolov et al., 2013a;b) and GloVe (Pennington et al., 2014) enabled state-of-the-art results on myriad NLP tasks such as sentiment analysis (Kim, 2014; Tai et al., 2015) and textual entailment (Chen et al., 2017).

However, for *polysemous* words (those with multiple senses), learning a single vector for each word type conflates different meanings (e.g., "A hydrogen bond exists between water molecules." vs. "Do you want to buy this bond?"). This is not a new problem—Schütze (1998) demonstrates the deficiency of assigning just one vector per word—but it is more pernicious in modern models, as conflated senses can pull semantically unrelated words toward each other in the embedding space (Neelakantan et al., 2014; Pilehvar & Collier, 2016; Camacho-Collados & Pilehvar, 2018). To disentangle distinct senses in word embeddings and learn finer-grained semantic clusters, *multi-prototype word embedding* models learn multiple sense-specific embeddings for a single word (Section 7).

But what makes a good multisense word embedding? While word similarity is the most common evaluation, it has many detractors (Faruqui et al., 2016; Gladkova & Drozd, 2016): similarity is subjective and is hard to be differentiate from word relatedness. Moreover, word similarity tasks—with the exception of Stanford Contextual Word Similarity (Huang et al., 2012, SCWS)—ignore polysemous cases or are tied to specific sense inventories (Boyd-Graber et al., 2006).

Moreover, these evaluations ignore a key component of learning sense inventories: do they make sense to a human? Previous multisense embedding papers present nearest neighbors to claim their representations are interpretable and useful. Like topic models, these implicit interpretability claims need to be rigorously verified. In Section 6, we adapt techniqes for evaluating topic models (Chang et al., 2009) to measure whether learned sense groups are internally coherent and whether humans can consistently match a learned sense vector to a word in context. Just like topic models, word embedding models that win conventional evaluations do not always make sense to humans.

We present a simple method that not only correlates well with traditional word similarity evaluations (Section 5) but also discovers interpretable (measured by human evaluations) sense embeddings (Section 6). Our model extends the Skip-Gram Word2Vec model and simultaneously learns (1) automatic sense induction given local context and (2) sense-specific embeddings. To learn disentangled sense representations (i.e., avoid sense mixing), we approximate hard attention and preserve

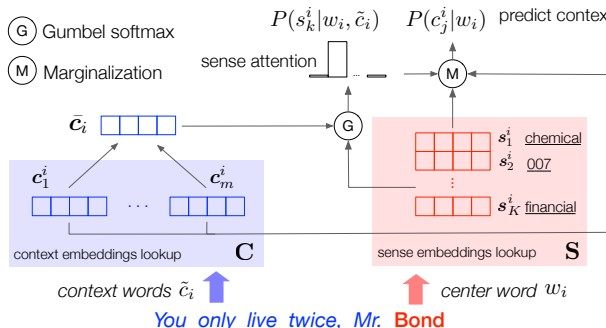

Figure 1: Network struture with an example of our GASI model which learns a set of global context embeddings $\mathbf{C}$ and a set of sense embeddings $\mathbf{S}$

differentiability via a scaled variant of the Gumbel Softmax function (Section 3.2). This modeling contribution—*Scaled Gumbel Softmax*—is critical for disambiguating senses.

## 2 FOUNDATIONS: SKIP-GRAM AND GUMBEL SOFTMAX

Our model extends Skip-Gram Word2Vec (Mikolov et al., 2013a;b), which jointly learns word embeddings $\mathbf{W} \in \mathbb{R}^{|V| \times d}$ and context embeddings $\mathbf{C} \in \mathbb{R}^{|V| \times d}$. More specifically, given a vocabulary $V$ and embedding dimension $d$, it maximizes the likelihood of the context words $c_j^i$ that surround a given center word $w_i$ in a context window $\tilde{c}_i$,

$$J(\mathbf{W}, \mathbf{C}) \propto \sum_{w_i \in V} \sum_{c_j^i \in \tilde{c}_i} \log P(c_j^i \,|\, w_i; \mathbf{W}, \mathbf{C}), \tag{1}$$

where $P(c_j^i \,|\, w_i)$ is estimated by a softmax over all possible context words, i.e, the vocabulary,

$$P(c_j^i \,|\, w_i; \mathbf{W}, \mathbf{C}) = \frac{\exp\left(\boldsymbol{c}_j^{i\top} \boldsymbol{w}_i\right)}{\sum_{c \in V} \exp\left(\boldsymbol{c}^\top \boldsymbol{w}_i\right)}. \tag{2}$$

In practice, $\log P(c_j^i \,|\, w_i)$ is approximated by negative sampling to reduce computational cost.

### 2.1 GUMBEL SOFTMAX

The Gumbel softmax (Jang et al., 2016; Maddison et al., 2016) approximates the sampling of discrete random variables. Given a discrete random variable $X$ with $P(X = k) \propto \alpha_k$, $\alpha_k \in (0, \infty)$, the Gumbel-max (Gumbel & Lieblein, 1954; Maddison et al., 2014) refactors the sampling of $X$ into

$$X = \arg\max_k (\log \alpha_k + g_k), \tag{3}$$

where the Gumbel noise $g_k = -\log(-\log(u_k))$ and $u_k$ are i.i.d samples drawn from Uniform(0, 1).

The Gumbel softmax approximates sampling one_hot($\arg\max_k(\log \alpha_k + g_k)$) by

$$y_k = \text{softmax}((\log \alpha_k + g_k)/\tau). \tag{4}$$

## 3 GUMBEL-ATTENTION SENSE INDUCTION (GASI)

Building on these foundations, we now introduce our model, GASI, and along the way introduce a soft-attention stepping-stone (SASI); afterward, we will compare models on both traditional evaluation metrics and interpretability. The critical component of our model is that we model the sense selection probability, which can be interpreted as sense attention over contexts, into the Skip-Gram model while preserving the original objective through marginalization (Figure 1). By using Gumbel Softmax, our model both approximates discrete sense selection and is differentiable. Previous models are either non-differentiable or otherwise complicate inference through hard attention with reinforcement learning methods (Lee & Chen, 2017).

### 3.1 ATTENTIONAL SENSE INDUCTION FRAMEWORK

**Embedding Parameters** We learn a context embedding matrix $\mathbf{C} \in \mathbb{R}^{|V| \times d}$ and a sense embedding tensor $\mathbf{S} \in \mathbb{R}^{|V| \times K \times d}$. Unlike previous work (Neelakantan et al., 2014; Lee & Chen, 2017), no extra embeddings are kept for sense induction.

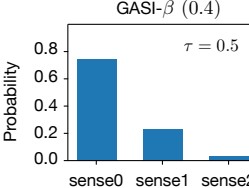 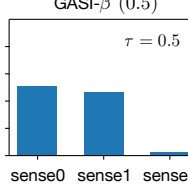 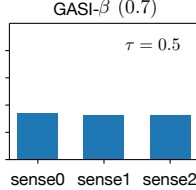 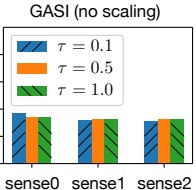

Figure 2: As the scale factor $\beta$ increases, the sense selection distribution for "bond" given examples from SemCor 3.0 for synset "bond.n.02" becomes flatter, indicating less disambiguated sense vectors.

**Number of Senses** For simplicity and consistency with most previous work, our model has a fixed number of senses $K$.[1]

**Sense Attention in Objective Function** Assuming a center word $w_i$ has senses $\{s_1^i, s_2^i, \ldots, s_K^i\}$, the original Skip-Gram likelihood can be written as marginal distribution over all senses of $w_i$ with the sense induction probability $P(s_k^i \,|\, w_i)$, we focus on the sense disambiguation given local context $\tilde{c}_i$ and estimate

$$P(c_j^i \,|\, w_i) = \sum_{k=1}^{K} P(c_j^i \,|\, s_k^i)P(s_k^i \,|\, w_i) \approx \sum_{k=1}^{K} P(c_j^i \,|\, s_k^i) \underbrace{P(s_k^i \,|\, w_i, \tilde{c}_i)}_{\text{attention}}, \tag{5}$$

Replacing $P(c_j^i \,|\, w_i)$ in Equation 1 with Equation 5 gives our objective function

$$J(\mathbf{S}, \mathbf{C}) \propto \sum_{w_i \in V} \sum_{c_j^i \in \tilde{c}_i} \log \sum_{k=1}^{K} P(c_j^i \,|\, s_k^i)P(s_k^i \,|\, w_i, \tilde{c}_i). \tag{6}$$

**Lower Bound the Objective for Negative Sampling** Like the Skip-Gram objective (Equation 2), we model the likelihood of a context word given the center sense $P(c_j^i \,|\, s_k^i)$ using softmax,

$$P(c_j^i \,|\, s_k^i) = \frac{\exp\left(\boldsymbol{c}_j^{i\top} \boldsymbol{s}_k^i\right)}{\sum_{j=1}^{|V|} \exp\left(\boldsymbol{c}_j^\top \boldsymbol{s}_k^i\right)}, \tag{7}$$

where the bold symbol $\boldsymbol{s}_k^i$ is the vector representation of sense $s_k^j$ from $\mathbf{S}$, and $\boldsymbol{c}_j$ is the context embedding of word $c_j$ from $\mathbf{C}$.

Computing the softmax over the vocabulary is time-consuming. We want to adopt negative sampling to approximate $\log P(c_j^i \,|\, s_k^i)$, which does not exist explicitly in our objective function (Equation 6).[2]

However, given the concavity of the logarithm function, we can apply Jensen's inequality,

$$\log \sum_{k=1}^{K} P(c_j^i \,|\, s_k^i)P(s_k^i \,|\, w_i, \tilde{c}_i) \geq \sum_{k=1}^{K} P(s_k^i \,|\, w_i, \tilde{c}_i) \log P(c_j^i \,|\, s_k^i), \tag{8}$$

and create a lower bound of the objective. Maximizing this lower bound gives us a *tractable objective*,

$$J(\mathbf{S}, \mathbf{C}) \propto \sum_{w_i \in V} \sum_{c_j^i \in \tilde{c}_i} \sum_{k=1}^{K} P(s_k^i \,|\, w_i, \tilde{c}_i) \log P(c_j^i \,|\, s_k^i), \tag{9}$$

where $\log P(c_j^i \,|\, s_k^i)$ is estimated by negative sampling Mikolov et al. (2013b),

$$\log \sigma(\boldsymbol{c}_j^{i\top} \boldsymbol{s}_k^i) + \sum_{j=1}^{n} \mathbb{E}_{c_j \sim P_n(c)}[\log \sigma(-\boldsymbol{c}_j^\top \boldsymbol{s}_k^j))], \tag{10}$$

---

[1]We can prune the duplicated senses for words that have senses less than $K$, details in Appendix B. We can also set different number of senses based on word frequency in the training, details in Appendix B.3.

[2]Deriving the negative sampling requires the logarithm of a softmax (Goldberg & Levy, 2014).

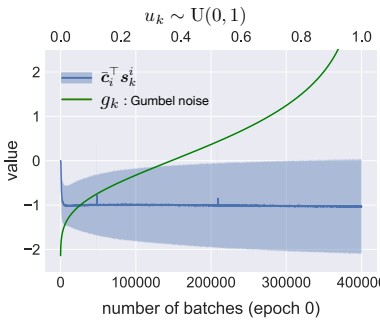

Figure 3: Our hard attention mechanism is approximated with Gumbel softmax on the context-sense dot product $\bar{c}_i^\top s_k^i$ (Equation 13), whose mean and std plotted here as a function of iteration. The shadowed area shows that it has a smaller scale than the gumbel noise $g_k$, such that $g_k$, rather than the embeddings, dominates the sense attention.

**Modeling Sense Attention**   We can model the attention term, *contextual sense induction distribution*, with soft attention; we call the resulting model soft-attention sense induction (SASI); although it is a stepping stone to our final model, we compare against it in our experiments as it helps isolate the contributions of hard attention. In SASI, the sense attention is conditioned on the entire local context $\tilde{c}_i$ with softmax:

$$P(s_k^i \mid w_i, \tilde{c}_i) = \frac{\exp\left(\bar{c}_i^\top s_k^i\right)}{\sum_{k=1}^{K} \exp\left(\bar{c}_i^\top s_k^i\right)}, \tag{11}$$

where $\bar{c}_i$ is the mean of the context vectors in $\tilde{c}_i$.

### 3.2   Scaled Gumbel Softmax for Sense Disambiguation

To reduce separate senses and learn *distinguishable sense representations*, we implement *hard* attention in our full model, GASI. To preserve differentiability and circumvent the difficulties in training with reinforcement learning (Sutton & Barto, 1998), we apply the reparameterization trick with Gumbel softmax (Section 2.1) to our sense attention function (Equation 11) and make a continuous relaxation.

**Vanilla Gumbel Attention**   The discrete sense sampling from Equation 11 can be refactored by

$$z^i = \text{one\_hot}(\arg\max_k(\bar{c}_i^\top s_k^i + g_k)), \tag{12}$$

and the hard attention is approximated with

$$y_k^i = \text{softmax}((\bar{c}_i^\top s_k^i + g_k)/\tau). \tag{13}$$

**Scaled Gumbel Softmax for Sense Disambiguation**   Gumbel softmax learns a flat distribution over senses even with low temperatures (Figure 2): the dot product $\bar{c}_i^\top s_k^i$ is too small compared to the Gumbel noise $g_k$ (Figure 3).[3] Thus we use a scaling factor $\beta$ to reduce the randomness,[4] and tune it as a hyperparameter.[5]

$$\gamma_k^i = \text{softmax}((\bar{c}_i^\top s_k^i + \beta g_k)/\tau), \tag{14}$$

We use GASI-$\beta$ to identify the GASI model with scaling factor. This modification is critical for learning *distinguishable senses* (Figure 2, Table 1, and Table 5).

**Final Objective Function**   The objective function of our GASI-$\beta$ model is

$$J(\mathbf{S}, \mathbf{C}) \propto \sum_{w_i \in V} \sum_{w_c \in c_i} \sum_{k=1}^{K} \text{softmax}((\bar{c}_i^\top s_k^j + \beta g_k)/\tau) \log P(w_c \mid s_k^i). \tag{15}$$

---

[3] Float32 precision, the saturation of $\log(\sigma(\cdot))$ and gradient vanishing result in a small range of $\bar{c}_i^\top s_k^i$.

[4] Normalizing $\bar{c}_i^\top s_k^i$ or directly using $\log P(s_k^i \mid w_i, \tilde{c}_i)$ results in a similar outcome.

[5] Learning $\beta$ instead of fixing it as a hyperparameter does not successfully disambiguate senses.

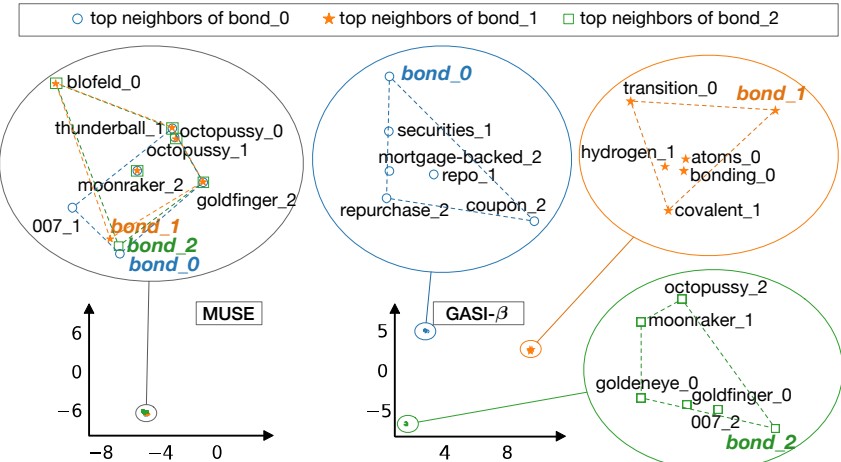

Figure 4: t-SNE projections of nearest neighbors for "bond" by *hard-attention* models: 1) previous SOTA model MUSE (RL-based); 2) our proposed GASI-$\beta$. Trained on same dataset and vocabulary, both models learn three vectors per word. Here, $word\_i$ represent the $i$-th vector for $word$. Our GASI (right) learns three distinct senses of "bond" while MUSE (left) learns overlapping senses.

## 4 TRAINING SETTINGS

For fair comparisons, we try to remain consistent with previous work (Huang et al., 2012; Neelakantan et al., 2014; Lee & Chen, 2017) in all aspects of training. In particular, we train GASI on the same April 2010 Wikipedia snapshot (Shaoul C., 2010) with 1B tokens the same vocabulary released by Neelakantan et al. (2014); set the number of senses $K = 3$ and dimension $d = 300$ for each word unless otherwise specified. More details are in Appendix A. We fix the temperature $\tau = 0.5$,[6] and tune the scaling factor $\beta$ from $\{0.1, 0.2, ...,0.9\}$ on the AvgSimC measure for the contextual word similarity task (Section 5). The optimal scaling factor $\beta$ is 0.4. If not reprinted, numbers for competing models are either computed with pre-trained embeddings released by authors or trained on released code.[7]

## 5 WORD SIMILARITY EVALUATION

We first compare our GASI and GASI-$\beta$ model with previous work on standard word similarity tasks before turning to interpretability experiments. Each task has word pairs with a similarity/relatedness score. For evaluation, we measure Spearman's rank correlation $\rho$ (Spearman, 1904) between word embedding similarity and the gold similarity judgements: higher scores imply the model captures semantic similarities consistent with the trusted similarity scores.

**Contextual Word Similarity**   Tailored for sense embedding evaluation, Stanford Contextual Word Similarities (Huang et al., 2012, SCWS) has 2003 word pairs and similarity scores with sentential context. Moreover, the word pairs and their contexts reflect homonymous and polysemous words. Therefore, we use this dataset to tune our hyperparameters.

To compute the word similarity with senses we use two metrics Reisinger & Mooney (2010) that take context and sense disambiguation into account: **MaxSimC** computes the cosine similarity $\cos(s_1^*, s_2^*)$ between the two most probable senses $s_1^*$ and $s_2^*$ that maximizes $P(s_k^i \mid w_i, \tilde{c}_i)$. **AvgSimC** weights average similarity over the combinations of all senses $\sum_{i=1}^{K} \sum_{i=j}^{K} P(s_i^1 \mid w_1, \tilde{c}_1) P(s_j^2 \mid w_2, \tilde{c}_2) \cos(s_i^1 s_j^2)$.

We compare variants of our model with multi-prototype sense embedding models (Table 1), including two previous state-of-the-art models: the clustering-based Multi-Sense Skip-Gram model (Neelakantan et al., 2014, MSSG) on AvgSimC metric and the RL-based Modularizing Unsupervised

---

[6]This is similar to the experiment settings for Gumbel softmax in Maddison et al. (2016)

[7]We adopt the numbers for Li & Jurafsky (2015) from Lee & Chen (2017) and tune the PDF-GM (Athiwaratkun et al., 2018) model on the same 1B corpus and vocabulary as previous works using https://github.com/benathi/multisense-prob-fasttext with suggested hyperparameters and select the best results.

| Model | MaxSimC | AvgSimC |
|---|---|---|
| Huang et al. (2012)-50d | 26.1 | 65.7 |
| MSSG-6K | 57.3 | 69.3 |
| MSSG-30K | 59.3 | 69.2 |
| Tian et al. (2014) | 63.6 | 65.4 |
| Li & Jurafsky (2015) | 66.6 | 66.8 |
| Qiu et al. (2016) | 64.9 | 66.1 |
| Bartunov et al. (2016) | 53.8 | 61.2 |
| MUSE_Boltzmann | 67.9 | 68.7 |
| SASI | 55.1 | 67.8 |
| GASI(w/o scaling) | **68.2** | 68.3 |
| GASI-$\beta$ | 66.4 | **69.5** |

Table 1: Spearman's correlation $100\rho$ on SCWS (trained 1B token, 300d vectors except for Huang et al.)

| Model | Accuracy(%) |
|---|---|
| *Unsupervised Multi-prototype models* | |
| MSSG-30K | 54.00 |
| MUSE_Boltzmann | 52.14 |
| GASI-$\beta$ | **55.27** |
| *Multi-prototype models with external lexical resources* | |
| DeConf | **58.55** |
| SW2V | 54.56 |

Table 2: Unsupervised sense selection accuracy on Word in Context

Sense Embeddings (Lee & Chen, 2017, MUSE) on MaxSimC. All three are better than the baseline Skip-Gram model (65.2 using the word embedding). GASI better captures similarity than SASI, corroborating that hard attention aids word sense selection. GASI without scaling ($\beta$) has the best MaxSimC; however, it learns a flat sense distribution (Figure 2). GASI-$\beta$ has the best AvgSimC and a competitive MaxSimC. While MUSE has a higher MaxSimC than GASI-$\beta$, it fails to distinguish senses as well (Figure 4, Section 6). The Probabilistic FastText Gaussian Mixture (Athiwaratkun et al., 2018, PDF-GM) is SOTA on multiple non-contextual word similarity tasks (Table 3). Without sense selection module given context, we evaluate PDF-GM on MaxSim (Equation 16), which is 66.4. Our GASI-$\beta$ has the same on MaxSim, and better correlation on AvgSimC (69.5).

**Word Sense Selection in Context** SCWS evaluates models' ability of sense selection indirectly. We further compare GASI-$\beta$ with previous SOTA, MSSG-30K and MUSE, on the Word in Context dataset (Pilehvar & Camacho-Collados, 2018, WiC) which requires the model to identify whether a word has the same sense in two contexts. Lacking ground truth for the development set,[8] to reduce the variance in training and to focus on evaluating the sense selection module, we use an evaluation suited for unsupervised models: if the model selects different sense vectors given contexts, we mark that the word has different senses.[9] For MUSE, MSSG and GASI-$\beta$, we use each model's sense selection module; for DeConf (Pilehvar & Collier, 2016) and SW2V (Mancini et al., 2017), we follow Pilehvar & Camacho-Collados (2018) and Pelevina et al. (2016) by selecting the closest sense vectors to the context vector. Results on DeConf are comparable to supervised results (59.4$\pm$ 0.7). Our GASI-$\beta$ has the best result apart from DeConf itself, which uses the same sense inventory (Miller & Fellbaum, 1998, WordNet) used to build WiC.

This evaluation, however, does not reflect the interpretability of the senses themselves. We address this in Section 6.

**Non-Contextual Word Similarity** To evaluate the semantics captured by each sense-specific embeddings, we compare the models on the non-contextual word similarity datasets: RG-65 (Rubenstein & Goodenough, 1965); SimLex-999 (Hill et al., 2015); WS-353 (Finkelstein et al., 2002); MEN-3k (Bruni et al., 2014); MC-30 (Miller & Charles, 1991); YP-130 (Yang & Powers, 2006); MTurk-287 (Radinsky et al., 2011); MTurk-771 (Halawi et al., 2012); RW-2k (Luong et al., 2013). Similar to Lee & Chen (2017) and Athiwaratkun et al. (2018), we compute the word similarity based on senses by **MaxSim** (Reisinger & Mooney, 2010), which maximizes the cosine similarity over the combination of all sense pairs and does not require local contexts,

$$\text{MaxSim}(w_1, w_2) = \max_{0 \le i \le K, 0 \le j \le K} \cos(s_i^1, s_j^2). \tag{16}$$

GASI-$\beta$ has better correlation on three datasets, is competitive on the rest (Table 3), and remains competitive without scaling. GASI is better than MUSE, the other hard-attention multi-prototype model, on six datasets and worse on three. Our model can reproduce word similarities as well or better than existing models through our sense selection.

---

[8]Unavailable as of November 2018 at `https://pilehvar.github.io/wic/`

[9]For words not in vocabulary or only have one sense learned, we chose randomly.

| Dataset | MSSG-30K | MSSG-6K | MUSE_Boltzmann | SASI | GASI | GASI-$\beta$ | PFT-GM |
|---------|----------|---------|----------------|------|------|--------------|--------|
| SimLex-999 | 31.80 | 28.65 | 39.61 | 31.56 | 40.14 | **41.68** | 40.19 |
| WS-353 | 65.69 | 67.42 | 68.41 | 58.31 | 68.49 | **69.36** | 68.6 |
| MEN-3k | 65.99 | 67.10 | 74.06 | 65.07 | 73.13 | 72.32 | **77.40** |
| MC-30 | 67.79 | 76.02 | 81.80 | 70.81 | 82.47 | **85.27** | 74.63 |
| RG-65 | 73.90 | 64.97 | **81.11** | 74.38 | 77.19 | 79.77 | 79.75 |
| YP-130 | 40.69 | 42.68 | 43.56 | 48.28 | 49.82 | 56.34 | **59.39** |
| MT-287 | 65.47 | 64.04 | 67.22 | 64.54 | 67.37 | 66.13 | **69.66** |
| MT-771 | 61.26 | 58.83 | 64.00 | 55.00 | 66.65 | 66.70 | **68.91** |
| RW-2k | 42.87 | 39.24 | **48.46** | 45.03 | 47.22 | 47.69 | 45.69 |

Table 3: Spearman's correlation $100\rho$ on non-contextual word similarity measured by MaxSim. GASI-$\beta$ outperforms the other models on three datasets are competitive on the others. Note that PFT-GM are trained with two components/senses while other models learn three senses.

## 6 CROWDSOURCING EVALUATION

GASI can capture word similarity (Section 5), but do the learned representations make sense? Could a human use them to help build a dictionary? If you show a human the senses, can they understand why a model would assign a sense to that context? In this section we evaluate whether the representations make sense to human consumers of multisense models.

**Qualitive analysis** Previous papers use nearest neighbors of a few examples to qualitatively argue that their models have captured meaningful senses of words. We also give an example in Figure 4, which provides an intuitive view on how the learned senses are clustered by visualizing the nearest neighbors of word "bond" using t-SNE projection (Maaten & Hinton, 2008). Our proposed model (right) disentangles the three sense of "bond" clearly and learns three distinct sense vectors.

However, the examples can be cherry-picked and lack standards. This problem also bedeviled topic modeling until the introduction of rigorous human evaluation (Chang et al., 2009). We adapt both aspects Chang *et al*'s evaluations: *word intrusion* (Schnabel et al., 2015) to evaluate whether individual senses are coherent and *topic intrusion*—rather sense intrusion in this case—to evaluate whether humans agree with models' sense assignments *in context*. Both crowdsourcing tasks collect human inputs on Figure-Eight. We compare our models with two previous state-of-the-art multi-prototype sense embeddings models that disambiguate senses given local context, i.e., MSSG (Neelakantan et al., 2014) and MUSE (Lee & Chen, 2017).[10]

### 6.1 WORD INTRUSION FOR SENSE COHERENCE

Schnabel et al. (2015) suggests a "good" word embedding should have coherent neighbors and evaluate coherence by *word intrusion*. They presents crowdworkers four words: three are close in embedding space while one of which is an "intruder". If the embedding makes sense, contributors will easily spot the word that "does not belong".

Similarly, we examine the coherence of ten nearest neighbors of senses in the *contextual word sense selection* task (Section 6.2) and replace one neighbor with an "intruder" (Figure 5). We generate three intruders for each sense and collect three judgements per intruder. We consider the "intruder" to be correctly selected if at least two judgements are correct.

| Model | Sense-level Accuracy | Judgement-level Accuracy | Aggrement |
|-------|----------------------|---------------------------|-----------|
| MUSE | 67.33 | 62.89 | 0.73 |
| MSSG-30K | 69.33 | 66.67 | 0.76 |
| GASI-$\beta$ | **71.33** | **67.33** | **0.77** |

Figure 5: Word intrusion task prompt

Table 4: Word intrusion evalutations on top ten nearest neighbors of sense embeddings.

---

[10]MSSG has two settings; we run human evaluation with MSSG-30K which has higher correlation with MaxSimC on SCWS.

Figure 6: An example (target: *bond*) of the *contextual word sense selection* task; each option contains top ten nearest neighbors of a sense embedding learned by the model; senses in this example are from our GASI-$\beta$ (1. 007; 2. chemical; 3. financial).

Like Chang et al. (2009), we want the "intruder" to not be too different in terms of frequency to the target set but not too similar semantically. For sense $s_i^m$ of word type $w_i$, we randomly select a word from the neighbors of another sense $s_i^n$ of $w_i$ but with a low threshold, i.e., any words that has cosine similarity larger than 0.0 can be viewed as a neighbor.

**Result and Analysis**   All models have comparable model accuracy. GASI-$\beta$ learns senses that have the highest coherency among top ten nearest neighbors while MUSE learns more sense mixtures.

**Inter-rater Agreement**   We use the aggregated confidence score provided by Figure-Eight to estimate the level of agreement between multiple contributors.[11] The agreements are high for all models and our GASI-$\beta$ has the highest agreement, suggesting that the senses learned by GASI-$\beta$ are easier to interpret.

## 6.2   CONTEXTUAL WORD SENSE SELECTION

The previous task measures whether individual senses are coherent. In this task, we measure whether the learned senses by sense embedding models make sense human and evaluate the models' ability to disambiguate senses in context.

**Task Description**   Given a target word in context, we ask a crowdworker to select which sense group best fits the sentence. Each sense group is described by its top ten distinct nearest neighbors (Figure 6).[12]

**Data Collection**   We select fifty nouns with five sentences from SemCor 3.0 (Miller et al., 1994). We first filter all word types with fewer than ten sentences and select the fifty most polysemous nouns from WordNet (Miller & Fellbaum, 1998) among the remaining senses. For each noun, we randomly select five sentences.

**Metrics**   For each model, we collect three judgements for each question. We consider a model correct if at least two crowdworkers select the same sense as the model. We also consider the probability $P$ assigned to the human choices by the model, indicating the model's confidence in sense selection. $P = 1/3$ indicates the model learns flat, uniform sense induction distribution is unable to disambiguate senses.

**Sense disambiguation and interpretability**   If humans consistently pick the same sense as the model: 1) humans can interpret the nearest neighbor words (as measured by the previous experiment); 2) the senses are distinguishable to human; 3) the human's choice is consistent with the model's.

**Results and Analysis**   GASI-$\beta$ selects senses that are most consistent with humans; it has the highest accuracy and assigns the largest probability assigned to the human choices (Table 5). Thus, GASI-$\beta$ produces sense embeddings that are both more interpretable and distinguishable. GASI without a scaling factor, however, has low consistency and flat sense distribution.

**Inter-rater Agreement**   We use the confidence score computed by Figure-Eight to estimate the rater's agreement for this task as well. Our GASI-$\beta$ achieves the highest human-model agreement while both MUSE and GASI without scaling have the lowest.

---

[11]https://success.figure-eight.com/hc/en-us/articles/201855939-How-to-Calculate-a-Confidence-Score

[12]We shuffle the choices for questions with the same target word.

| Model | Accuracy | $P$ | Agreement |
|---|---|---|---|
| MUSE | 28.0 | 0.33 | 0.68 |
| MSSG-30K | 44.5 | 0.37 | 0.73 |
| GASI (no $\beta$) | 33.8 | 0.33 | 0.68 |
| GASI-$\beta$ | **50.0** | **0.48** | **0.75** |

| | | MUSE | MSSG | GASI-$\beta$ |
|---|---|---|---|---|
| word overlaps | correct | 4.78 | 0.39 | 1.52 |
| | error | 5.43 | 0.98 | 6.36 |
| cosine_sim by Glove | correct | 0.86 | 0.33 | 0.36 |
| | error | 0.88 | 0.57 | 0.81 |

Table 5: Human-model consistency on *contextual word sense selection*; $P$ is the average probability assigned by the model to the human choices. GASI-$\beta$ is most consistent with human.

Table 6: Similarities of human and model choices when they disagree (error) vs. similarities between the senses that both human and model select with other senses in the same word (correct). Human agrees with the model when the senses are distinct.

**Error Analysis**  Next, we attempt to answer why crowdworkers disagree with the model although they can interpret most senses (measured by the word intruder task, Table 4). Is it that the model has learned *duplicate* senses that both the users and model cannot distinguish or is it that crowdworkers agree with each other but *disagree* with the model? The former relates to the model's ability in learning human distinguishable senses; while the latter relates to the model's ability in contextual sense selection.

Two trends reveal that duplicated senses that are not distinguishable to humans are one of the main causes of human-model disagreement. First, users agree with the model when the senses are distinct (Table 6, correct), while disagreement rises with more similar senses (Table 6, error); second, more distinct senses allows higher inter-rater agreement (Figure 7). We measure distinctness both by counting the number of shared nearest neighbors and the average cosine simlarities of GloVe embeddings.[13] Specifically, MUSE learns duplicate senses for most words, preventing users from choosing appropriate senses and results in random human-model agreement. GASI-$\beta$ learns some duplicated senses and some distinguishable senses. MSSG appears to learn the least similar senses, but they are not distinguishable enough for humans. For MSSG, small neighbor overlaps do not necessarily help humans to distinguish between senses: users disagree with each other (agreement 0.33) even when the number of overlaps is very small (Figure 7). An intuitive example is shown in Table 7, which demonstrates the necessity of human evaluation. If we use rater agreement to measure how distinguishable the learned senses are to humans, GASI-$\beta$ learns the most distinguishable senses (histogram in Figure 8).

Figure 8 also shows that the model is more likely to agree with humans when humans agree more with each other (as a result of more distinct senses), i.e., human-model consistency correlates with rater agreement (Figure 8). MSSG disagrees with humans more even when raters agree with each other, indicating worse sense selection ability.

### 6.3 WORD SIMILARITY VS. SENSE DISAMBIGUATION

The evaluation results on word similarity tasks (Section 5) and human evaluations (Section 6) are inconsistent for several models. GASI, GASI-$\beta$ model and the MUSE model are all competitive in word similarity (Table 1 and Table 3), but only GASI-$\beta$ also does well in the human evaluations (Table 5). Both GASI without scaling and MUSE fail to learn distinguishable senses and cannot disambiguate senses given local context. High word similarities do not necessarily indicate "good" sense embeddings quality; our human evaluation—*contextual word sense selection*—is complementary.

### 7 RELATED WORK

Schütze (1998) introduces context-group discrimination for senses and uses the centroid of context vectors as a sense representation. Other work induces senses by context clustering (Purandare & Pedersen, 2004) or probabilistic mixture models (Brody & Lapata, 2009). Reisinger & Mooney (2010) first introduce multiple sense-specific vectors for each word, inspiring other multi-prototype sense embedding models. Generally, to address polysemy in word embeddings, some previous work

---

[13]Different models learn different representations; we use GloVe for a uniform basis of comparison.

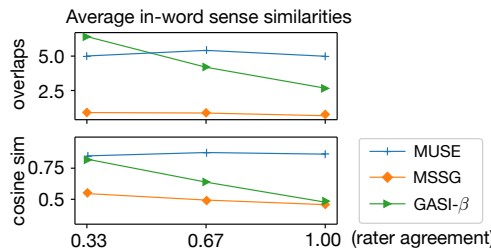

Figure 7: More distinct senses within each word lead to higher inter-rater agreement

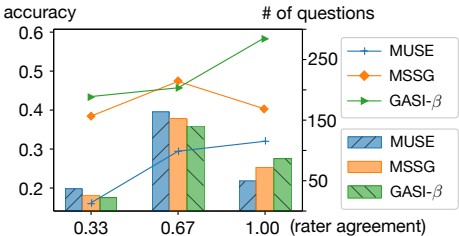

Figure 8: Higher inter-rater agreement correlates with higher human-model consistency.

trained on annotated sense corpora (Iacobacci et al., 2015) or external sense inventories (Labutov & Lipson, 2013; Chen et al., 2014; Jauhar et al., 2015; Chen et al., 2015; Wu & Giles, 2015; Pilehvar & Collier, 2016; Mancini et al., 2017); Rothe & Schütze (2015; 2017) extend word embeddings to lexical resources without training; others induce senses via multilingual parallel corpora (Guo et al., 2014; Šuster et al., 2016; Ettinger et al., 2016).

We contrast our GASI to unsupervised monolingual multi-prototype models along two dimensions: *sense induction methodology* and *differentiability*. 1), Huang et al. (2012) and Neelakantan et al. (2014) induce senses by context clustering; Tian et al. (2014) model a corpus-level sense distribution; Li & Jurafsky (2015) model the sense assignment as a Chinese Restaurant Process; Qiu et al. (2016) induce senses by minimizing an energy function on a context-depend network; Bartunov et al. (2016) model the sense assignment as a steak-breaking process; Nguyen et al. (2017) model the sense embeddings as a weighted combination of topic vectors with pre-computed weights by topic models; Athiwaratkun & Wilson (2017) and Athiwaratkun et al. (2018) model word representations as Gaussian Mixture embeddings where each Gaussian component captures different senses; Lee & Chen (2017) computes sense distribution by a separate set of sense induction vectors; while our GASI marginalizes the likelihood of contexts over senses and induces senses by local context vectors; the most similar sense selection module is a bilingual model (Šuster et al., 2016) except that it does not introduce lower bound for negative sampling but uses weighted embeddings, which results in more sense mixture. 2), most sense selection models are *non-differentiable* and discretely select senses, with two exceptions: Šuster et al. (2016) use weighted vectors over senses; Lee & Chen (2017) implement hard attention with RL to mitigate the non-differentiability. In contrast, GASI keeps full differentiability by reparameterization and approximates discrete sense sampling with scaled Gumbel softmax.

# 8    CONCLUSION

The goal of multi-sense word embeddings is not just to win word sense evaluation datasets; rather, they should also *describe* language: given millions of tokens of a language, what are the patterns in the language that can help a lexicographer or linguist in day-to-day tasks like building dictionaries or understanding semantic drift. Our differentiable Gumbel Attention Sense Induction (GASI) offers a best of both worlds: comparable word similarities while also learning more distinguishable, interpretable senses.

| The real question is - how are those four years used and what is their value as training? |
|---|
| **MSSG**    s1: hypothetical, unanswered, topic, answered, discussion, yes/no, answer, facts
s2: toss-up, answers, guess, why, answer, trivia, caller, wondering, answering
s3: argument, contentious, unresolved, concerning, matter, regarding, debated, legality |

Table 7: A case where MSSG has low overlaps but confuses raters (agreement 0.33); model choses s1.

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

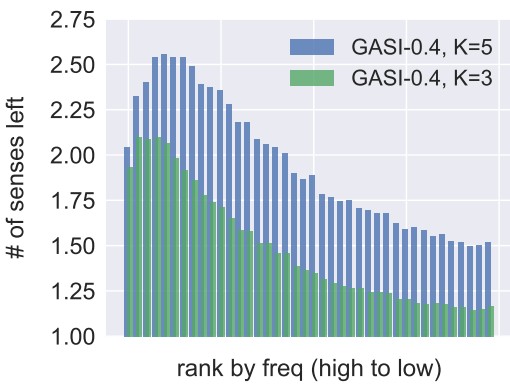

Figure 9: Histogram of number of senses left after post-training pruning for two models: GASI-0.4 initialized with three senses and GASI-0.4 initialized with five senses. We rank the number of senses of words by their frequency from high to low.

## A  TRAINING DETAILS

During training, we fix the window size to five and the dimensionality of the embedding space to 300 for comparison to previous work. We initialize both sense and context embeddings randomly within U(-0.5/dim, 0.5/dim) as in Word2Vec. We set the initial learning rate to 0.01; it is decreased linearly until training concludes after 5 epochs. The batch size is 512, and we use five negative samples per center word-context pair as suggested by Mikolov et al. (2013a). The subsample threshold is 1e-4. We train our model on the GeForce GTX 1080 Ti, and our implementation (using pytorch 3.0) takes $\sim 6$ hours to train one epoch on the April 2010 Wikipedia snapshot Shaoul C. (2010) with 100k vocabulary. For comparison, our implementation of Skip-Gram on the same framework takes $\sim 2$ hours each epoch.

## B  NUMBER OF SENSES

For simplicity and consistency with most of previous work, we present our model with a fixed number of senses $K$.

### B.1  POST-TRAINING PRUNING

For words that do not have multiple senses or have most senses appear very low-frequently in corpus, our model (as well as many previous models) learns duplicate senses. We can easily remove such duplicates by *pruning* the learned sense embeddings with a threshold $\lambda$. Specifically, for each word $w_i$, if the cosine distance between any of its sense embeddings $(s_m^i, s_n^i)$ is smaller than $\lambda$, we consider them to be duplicates. After discovering all duplicate pairs, we start pruning with the sense $s_k^i$ that has the most duplications and keep pruning with the same strategy until no more duplicates remain.

**Model-specific pruning**  We estimate a model-specific threshold $\lambda$ from the learned embeddings instead of deciding it arbitrary. Therefore, this pruning methods is also applicable to other sense embedding models. We first sample 100 words from the negative sampling distribution over the vocabulary. Then, we retrieve the five nearest neighbors (from all senses of all words) to each sense of each sampled word. If one of a word's own senses appears as a nearest neighbor, we append the distance between them to a *sense duplication list* $D_{dup}$. For other nearest neighbors, we append their distances to the *word neighbor list* $D_{nn}$. After populating the two lists, we want to choose a threshold that would prune away all of the sense duplicates while differentiating sense duplications with other distinct neighbor words. Thus, we compute

$$\lambda = \frac{1}{2}(\text{mean}(D_{dup}) + \text{mean}(D_{nn})). \tag{17}$$

| Model | MaxSimC | AvgSimC |
|---|---|---|
| GASI-0.4 | 66.4 | 69.5 |
| GASI-0.4-30K | 65.3 | 69.2 |
| GASI-0.4-post-pruning | 65.6 | 68.7 |

Table 8: Spearman's ranking correlation $100 \times \rho$ on SCWS. GASI-0.4-30K means top 30,000 words are initialized with three senses while the others have one sense.

Table 1 compares the sense embeddings after pruning with the original mode on the Stanford Contextual Word Similarities (SCWS) task Huang et al. (2012). Both AvgSimC and MaxSimC with post-pruning embeddings decrease only a few compare to that from GASI-0.4.

### B.2 NUMBER OF SENSES VS. WORD FREQUENCY

It is a common assumption that more frequent words have more senses. Figure 1 shows a histogram of the number of senses left for words ranked by their frequency, and the results agree with the assumption. Generally, the model learns more sense for high frequent words, except for the most frequent ones. The most frequent words are usually considered stopwords, such as "the", "a" and "our', which have only one common meaning. Moreover, we compare our model initialized with three senses (GASI-0.4, $K = 3$) against the one that has five (GASI-0.4, $K = 5$). Initializing with a larger number of senses, the model is able to uncover more senses for most words.

### B.3 INITIALZING $K$ BASED ON WORD FREQUENCY

Despite our model has a fixed number of senses. It is easy to implement our model with different number of senses with a mask matrix. And we can define different number of senses for each word based on their frequency. In Table 1, we show the results from a model that only top 30,000 word are initialized with three senses while others have one. The same choice is applied by Neelakantan et al. (2014).

