# OpenReview forum: "A Differentiable Self-disambiguated Sense Embedding Model via Scaled Gumbel Softmax"
_ICLR.cc/2019/Conference_

### Official Review · AnonReviewer3 · 2018-11-02
**This paper points out an important evaluation perspective, but the model architecture is incremental (limited novelty).**

**Rating:** 6
**Confidence:** 5

**Review:**

This paper proposes GASI to disambiguate different sense identities and learn sense representations given contextual information.
The main idea is to use scaled Gumbel softmax as the sense selection method instead of soft or hard attention, which is the novelty and contribution of this paper.
In addition, the authors proposed a new evaluation task, contextual word sense selection, which can be used to quantitatively evaluate the semantic meaningfulness of sense embeddings.
The proposed model achieves comparable performance on traditional word/sense intrinsic evaluation and word intrusion test as previous models, while it outperforms baselines on the proposed contextual word sense selection task.

While the scaled Gumbel softmax is the claimed novelty, it is more like an extension of the original MUSE model (Lee and Chen, 2017), which proposed the sense selection and representation learning modules for learning sense-level embeddings.
The only difference between the proposed one and Lee and Chen (2017) is Gumbel softmax instead of reinforcement learning between sense selection and representation learning modules.
Therefore, the idea from the proposed model is similar to Li and Jurafsky (2015), because the sense selection is not one-hot but a distribution.
The novelty of this paper is limited because the model is relatively incremental.

From my perspective, the more influential contribution is that this paper points out the importance of evaluating sense selection capability, which is ignored by most prior work.
Therefore, I expect to see more detailed evaluation on the selection module of the model.
Also, because the task of this paper is multi-sense embeddings, the traditional word similarity (without contexts) task seems unnecessary.
Moreover, there is no error analysis about the result on the proposed contextual word sense selection task, which may shed more light on the strength and weakness of the model.
Finally, I suggest the authors remove the word-level similarity task and try the recently released Word in Context (WiC) dataset, which is a binary classification task that determines whether the meaning of a word is different given two contexts.
It would be better to see that GASI performs well on this task given its better sense selection module.

Overall, the contribution is somewhat incremental and the evaluation/discussion should focus more on the sense selection module.
Considering the issues mentioned above, I will expect better quality for an ICLR paper.

---

> ### Author Response · Authors · 2018-11-25
> **Response to Reviewer 3**
>
> We thank the reviewer for taking time to read our paper and the useful suggestions! We address the reviewer’s concerns and suggestions as follows:
>
> 1) Additional evaluation with recently released WiC dataset
> >>>
> Finally, I suggest the authors remove the word-level similarity task and try the recently released Word in Context (WiC) dataset
> <<<
> We thank the reviewer for their suggestion! We add an evaluation on the recently released WiC dataset in the revision. We focus the evaluation on the sense selection module of the model and classify the senses in an unsupervised fashion. Our model achieves the highest accuracy among competing models (Table 2), except for DeConf which is a supervised sense model that annotates senses on the same lexical resource (WordNet) that was used to build WiC.
>
> We believe that the word similarity tasks demonstrate that each sense-specific embedding learned by our model captures good semantics in addition to better sense disambiguation ability. The high quality of each sense embeddings demonstrates the benefits of using Gumbel softmax. Therefore we decide to keep this evaluation in the revision, but it is more meaningful alongside the WiC results.
>
> 2) Novelty
>
> >>>
> The only difference between the proposed one and Lee and Chen (2017) is Gumbel softmax instead of reinforcement learning between sense selection and representation learning modules.
> <<<
> We appreciate that the reviewer noticed the similarity between our models with MUSE by Lee and Chen (2017), as both try to improve the sense selection module with hard attention. However, the overall structure of our model (Figure 1) is quite different, in addition to using Gumbel Softmax (GS) instead of RL for hard attention, we’d like to explain the differences on two key aspects:
>
> Model structure and parameters: MUSE learns *four* sets of parameters: sense representations for target words U, collocation context representations V, and two additional matrix P, Q to estimate sense selection distribution for both target words and contexts (both target and context have multiple senses).
> In contrast, ours learns *two* sets of parameters (Section 3.1): sense representations for target words S and global context representations C. We use C to disambiguate senses of target words S instead of using additional parameters like in MUSE, which reduces the number of total parameters in our model. Furthermore, we update S and C in both the sense selection and context prediction modules, as these two modules are “symmetric” (predicting senses by context and predict context by word sense) and both help to capture the semantics in words.  Moreover, similar to Neelakantan et al. (2014), we do not disambiguate senses for context words (one global vector per context word) to further reduce the parameter size.
>
> Optimization function: We use the (scaled) GS instead of straight-through (scaled) GS to have a stronger error signal (update not only the senses that are chosen but also the ones that are not).  To use a distribution instead of a one-hot selection and reduce the computational cost by negative sampling, we optimize the lower bound of the original negative sampling Skip-Gram objective with marginalization and Jensen's Inequality. Using straight-through (scaled) GS learns worse sense embeddings (lower word similarity score and human-model consistency) than (scaled) GS. Due to space limitations, we didn’t include the comparison in the paper. RL methods are similar to ST-GS since they also make a hard selection each time and update the selected senses but not others.
>
> >>>
> the idea from the proposed model is similar to Li and Jurafsky (2015), because the sense selection is not one-hot but a distribution.
> <<<
> Li and Jurafsky (2015) sample one-hot senses during the training with Chinese Restaurant Process (CRP) and model the CRP with a distribution; while we directly use the distribution and implement the standard skip-gram objective with marginalization over senses.
>
> 3) Error analysis
>
> >>>
> Moreover, there is no error analysis about the result on the proposed contextual word sense selection task, which may shed more light on the strength and weakness of the model.
> <<<
> We appreciate the reviewer’s suggestion! We add the error analysis on the crowdsourced contextual word sense selection task in the revision (Section 6.2 Error Analysis).

---

> > ### Comment · AnonReviewer3 · 2018-11-28
> > **Complete experiments and analysis**
> >
> > After seeing the author responses, my score is updated.
> >
> > The revised paper includes more experiments (almost all I can think of) and detailed analysis.
> > The difference between the proposed model and the prior work can be better elaborated for explicitly pointing out the novelty.
> > Also, the improved performance seems convincing for the proposed model, and it will be better to see the published code for encouraging researchers to easily follow up this direction.
> >
> > To sum up, this is a good paper that can motivate the following research in the related field.

---

> > > ### Author Response · Authors · 2018-11-28
> > > **Will release the code and data soon**
> > >
> > > We thank the reviewer for the new comments! We'll elaborate on the differences between our model with prior works with a little more detail in the next revision after the decision been made. And we'll release our code and data for the human evaluation as soon as possible.

---

### Official Review · AnonReviewer2 · 2018-11-05
**Interesting paper, promising results**

**Rating:** 6
**Confidence:** 4

**Review:**

* Summary

  This paper extends the skipgram model using one vector per sense of a word. Based on this, the paper proposes two models for training sense embeddings: One where the word senses are marginalized out with attention over the senses, and the second where only the sense with highest value of attention contributes to the loss. For the latter case, the paper uses a variant of Gumbel softmax for training. The paper shows evaluations on benchmark datasets that shows that the Gumbel softmax based method is competitive or better than other methods. Via a crowdsourced evaluation, the paper shows that the method also produces human interpretable clusters.

* Review
  This paper is generally well written and presents a plausible solution for the problem of discovering senses in an unsupervised fashion.

  If \beta=0, then we get SASI, right? How well does this perform on the non-contextual word similarity task? Also, on the crowd sourced evaluation? The motivation for the hard attention/Gumbel softmax is to learn sense representations that are distinguishable. But do the experiments test this?

  There's something strange about Eq 6. If I understand this correctly, \tilde{c_i} is the context and c_j^i is the j^th context word. Then P(c_j^i | w, \tilde{c_i}) should be 1 because the context is given, right? While the motivation for the right hand side makes sense, the notation could use work.

  The description of how the number of senses is pruned in section 3.1 seems to be a bit of a non sequitur. It is not clear whether this is used in the experiments and if so, how it compares. The appendix gives more details, but it seems a bit out of place even then because the evaluations don't seem to use it.


* Minor comments
  There are some places where the writing could be cleaned up.
  - Eq 16 changes the notation for the sense embeddings and the context words from earlier, say Eq 12.
  - Parenthetical citations would be more appropriate in some places Eg: above Eq 3, in footnote 3
  - Page 6, above 6.2: Figure-Figure?
  - Page 9, Agreement paragraph: hight -> highest

---

> ### Author Response · Authors · 2018-11-25
> **Response to Reviewer 2**
>
> We thank the reviewer for taking time to read our paper and the useful suggestions on improving our writing! We address the specific points from the reviewer as follows:
>
> >>>
> If \beta=0, then we get SASI, right? How well does this perform on the non-contextual word similarity task? Also, on the crowdsourced evaluation?
> <<<
> We thank the reviewer for this suggestion! We’ve added the results in Table 3 in the revision. SASI generally performs poorly on the word similarity tasks, so we focus our comparison between our main model GASI-beta with the baseline models given limited space.
>
> >>>
> The motivation for the hard attention/Gumbel softmax is to learn sense representations that are distinguishable. But do the experiments test this?
> <<<
>
> Our crowdsourced contextual sense selection task evaluates this property. The raters need to distinguish between the learned senses in order to make a selection (Section 6.2, sense disambiguation and interpretability). We also add more detail to these experiments in the additional error analysis in the revision.
>
> >>>
> There's something strange about Eq 6. ……,  While the motivation for the right hand side makes sense, the notation could use work.
> <<<
> We address the notation issue in the revision.
>
> >>>
> The description of how the number of senses is pruned in section 3.1 seems to be a bit of a non sequitur.
> <<<
> We thank the reviewer’s suggestion. Since it’s not the focus of our paper, in our revision we move the descriptions of pruning to the appendix.

---

### Official Review · AnonReviewer1 · 2018-11-05
**Neat idea applying Gumbel-softmax to multi sense embeddings**

**Rating:** 7
**Confidence:** 3

**Review:**

The paper presents a method for deriving multi sense word embeddings. The key idea behind this method is to learn a sense embedding tensor using a skip-gram style training objective. The objective defines the probability of contexts marginalised over latent sense embeddings. The paper uses Gumbel-softmax reparametrization trick to approximate sampling from the discrete sense distributions. The method also uses a separate hyperparameter to help scale the dot product appropriately.

Strengths:

1. The technique is a well-motivated solution for a hard problem that builds on the skip-gram model for learning word embeddings.
2. A new manual evaluation approach for comparing sense induction approaches.
3. The empirical advance while relatively modest appears to be significant since the technique seems to yield better results than multiple baselines across a range of tasks.

Suggestions:

1. The number of senses is fixed to three. This is a bit arbitrary, even though it is following some precedence. I like the information in the appendix that shows how to handle cases when there are duplicate senses induced for words that dont have many senses. It would be useful to know how to handle the cases where a word can have more than three senses. Given that the authors have a way of pruning duplicate senses, it would have been interesting to try a few basic methods that select the number of senses per word dynamically.

2. The evaluation includes word similarity task and crowdsourcing for sense intrusion and sense selection. These provide a measure of intrinsic quality of the sense based embeddings. However, as Li and Jurafsky (2015) point out, typically applications use more powerful models that use a wide context. It is not clear how these improvements to sense embeddings will translate in these settings. It would have been useful to have at least one or two end applications to illustrate this.


3. Given that the empirical gains are not quite consistent, I would encourage the authors to specifically argue why this particular method should be favoured over other existing methods. The related work discussion merely highlights methodological differences. For example, the contrast with Lee and Chen (2017) seems to be only that of differentiability. Is the claim that differentiability is desirable because this allows for fine tuning in applications? If this is the case then it will be nice to have this verified.

4. The lower bound on the log likelihood objective is good but what are we supposed to take away from it? Is it that there is an interpretation that allows us to get away with negative sampling?

Overall I like the paper. It presents an application of the Gumbel-softmax trick for sense embeddings induction and shows some empirical evidence for the usefulness of this idea, including some manual evaluation.

I think the evaluation could be strengthened with some end applications and much crisper arguments on why the method is preferable over other methods that achieve comparable performance.

References:

[Li and Jurafsky., EMNLP 2015] Do Multi-Sense Embeddings Improve Natural Language Understanding?

---

> ### Author Response · Authors · 2018-11-25
> **Response to Reviewer 1**
>
> We thank the reviewer for taking time to read our paper and the useful suggestions! We address the suggestions from the reviewer as follows:
>
> 1）dynamic number of senses and evaluation on downstream applications.
>
> We thank the reviewer for these two suggestions! The simplest way to model words that have more than 3 senses is to initialize all words with more senses and prune aggressively; we set K=3 mainly for purpose of comparison. We think both implementing a dynamic number of senses (e.g., by setting a threshold to split senses) and evaluating on end tasks are great ideas; given the limited space, we’ll address these in future work.
>
> 2）benefits of differentiability
>
> In addition to updating the sense selection module and context prediction module at the same time, full differentiability allows updates to flow to all senses, not only the ones chosen by the attention, which results in stronger error signals and better sense selection ability. While approximating hard attention still guarantees that the model will focus on specific senses so that each sense captures good semantics (Table 3) and is interpretable to humans (Section 6), the Gumbel-softmax trick helps to guarantee both with the original objective, and we don’t need additional parameters for the policy network in RL.
>
> >>>
> For example, the contrast with Lee and Chen (2017) seems to be only that of differentiability.
> <<<
> We also contrast the sense selection module with Lee and Chen (2017) in the related work and Section 3.1. The overall structure of the two models are actually different, Lee and Chen (2017) learn *four* set of parameters while we learn *two*. Given limited space, we don’t elaborate further in our paper, but we discuss this with more details in our response to Reviewer 3.
>
> 3) negative sampling and lower bound
>
> The negative sampling (NS) is for reducing the computational cost. To still optimize our original objective while implementing NS, we deduce the lower bound by Jensen’s Inequality.

---

### Public Comment · (anonymous) · 2018-12-01
**Some questions**

This is a nice work that extends current works to gumbel softmax and provides new experiment settings. However, I have some concerns based on my understanding. I hope that they can be answered by the author.

1. First of all, I have a methodological concern about the central assumption of the method Eq. (5). If I understand it correctly, the authors do not mention the conditional independence assumption (w -> s -> c) but instead using it directly. It suggests that the word w_i and the context words are conditionally independent given the underlying sense s^i_k. Isn't it true that the concrete realization of a sense (i.e., a word) drastically affect the context words? For example, "guy" and "man" are synonym, but one is more casual and the other is more formal. Hence, despite the same sense, different realization would cause the contexts to be more (or less) formal. One potential hope for this concern is that the sense of different words are not shared in your models, but given the premise is to learn embeddings of words such that the words with the same sense would have "the same" sense embeddings, such modeling framework does not solve this problem. In contrast, the recent trend to model the collocation of senses (Qiu et al., 2016, Lee & Chen, 2017) seems to avoid this problem. Why do the authors pursue an older approach (akin to (Li & Jurafsky, 2015)) that models the collocation of words and sense?

2. Also, the second "approximation" in Eq. (5) is very weird, how can p(s^i_k|w_i, \tilde{c}_i) be similar to p(s^i_k|w_i)? In the non-parametric sense, the author change the conditional table of (# of sense, # of words) for p(s^i_k|w_i) to (# of sense, # of words, # of contexts) for p(s^i_k|w_i, \tilde{c}_i), while the # of contexts are exponential to the # of words. It suggests a exponential difference in the complexity for the "approximation". (Sorry for that I cannot follow the remaining part of the methodology quite well because the remaining methods highly depend on the above assumptions.)

3. Why do the authors think modeling *two* sets of parameters (words and senses) as a novelty? Isn't it simply the conventional design as (Li & Jurafsky, 2015) that is different from more recent approaches that models a purely sense-based framework (Qiu et al., 2016, Lee & Chen, 2017)?

4. The gumbel softmax is a nice estimator that enables differentiability, but suffer from an biased gradient. The RL approach used by (Lee & Chen, 2017) provides unbiased gradient but suffer from large variance. Both approaches have their advantages and disadvantages, so I'm interested in seeing an ablation study that analyze the difference of gradient estimator in this task in terms of the impact of gradient estimation on the learning process. If the gumbel softmax estimation is an important novelty to this paper, the analysis of the estimator should be shown instead of the end-to-end performance.

---

> ### Author Response · Authors · 2018-12-03
> **Address the commenter's question 2, 3 and 4**
>
>
> 2) Approximation in Eq. (5)
>
> We add the approximation notation because we estimate the sense disambiguation distribution with local context instead of using the global word sense distribution (Tian et al. 2014). We assume that we can disambiguate senses based on local context. The same assumption is adopted by Neelakantan et al. (2014), Li and Jurafsky (2015), Lee and Chen (2017), etc. Moreover, if we consider the original Skip-Gram, each word-context pair counted is in a fixed context window, the local context is given, the left side is also implicitly conditioned on the local context within the window. We will revise the notation to make this clear.
>
>
> 3) Parameters
>
> Thanks for pointing out this. We do not claim in the paper that learning two sets of parameters is our contribution. In fact, several previous works apply the same parameter setting, such as Tian et al. (2014), Neelakantan et al. (2014) [the additional centers are computed with context parameters] and Li and Jurafsky (2015). Despite the same parameter setting, similar to previous work, our novelty lies in *different* mechanisms to disambiguate and select senses. We highlight the # of parameters in responses mainly to differentiate our work from Lee and Chen (2017), mentioning the parameter setting is also the basis in discussing sense disambiguation and sense selection mechanism.
>
>
> 4) Analysis of estimator
>
> >>>
> If the gumbel softmax estimation is an important novelty to this paper, the analysis of the estimator should be shown instead of the end-to-end performance.
> <<<
>
> We thank the commenter for this insight! We agree that Gumbel Softmax (GS) and RL approach have their advantages/disadvantages. However, the focus of our paper is not a direct comparison of GS and RL but rather to develop an efficient but effective sense embedding model and to answer “what are good sense embeddings”. The goal is to learn sense vectors that both capture semantics and are distinguishable to human. Thus, with limited space, following previous work, we focus on evaluating the quality of the learned embeddings and propose a new human evaluation method to exam the ability of the proposed model in learning human distinguishable senses. Moreover, in order to show the benefit of the proposed scaled GS, we perform the ablation study by comparing SASI, GASI and GASI-\beta.

---

> ### Author Response · Authors · 2018-12-03
> **Address the commenter's question 1**
>
> We thank the commenter for their interest in our paper and the valuable comments! We address the concerns as follows:
>
> 1) Model assumptions
>
> If we understand correctly, the commenter worries that the implied assumption w->s->c (first select a sense s from word w, then generate the context c) of our model is flawed, because they think w and c are not independent given s. They provide an example to show when w and c should not be independent given s. They suggest that we should model “the collocation of senses” (Qiu et al. 2016) instead of “the collocation of words and sense” like Li & Jurafsky (2015) to address this concern.
>
> First, we’d like to argue that no model assumption is perfect (https://en.wikipedia.org/wiki/All_models_are_wrong), for example, both Bag-of-Word model and LDA are flawed language model, it doesn’t mean that they’re not useful. Without mentioning what consequences our model assumptions would cause in training, testing, or evaluation, the commenter's concern seems to be purely cognitive/theoretical. Nevertheless, we're happy to discuss with the commenter about it
>
> We think that the commenter may have misunderstood a few points and the assumption w->s->c is not a concern. The conditional independence of w and c given sense could hold based on our assumption (we discuss using the commenter’s example in 1.2 below).
>
>
> Moreover, the evaluations demonstrate that our model is able to learn distinguishable senses and succeeds in both similarity-based and human evaluations. We agree that the model approach of Qiu et al. (2016) also makes sense, but it is not necessarily better than ours, as shown in our evaluation where ours outperform theirs (Table 1).
>
> In detail:
> 1.1 The commenter may have an inaccurate understanding about our premise.
>
> We’d like to reassure the commenter that our model does not share senses among words, like *all* multi-prototype models. Also, the premise is not to learn “same” embeddings for words that have the same sense. It is to learn similar embeddings for certain senses of words that are synonyms (share similar contexts). Two words may be synonyms, but they won’t have identical senses.
>
> 1.2  The commenter worries “c is not independent of w given s”, if we understand correctly, they think “not sharing senses” could be a solution but not quite.
>
> However, no matter the senses are shared across words or not, we can have c independent of w given s.
>
> A. When not sharing senses (our case),
>
>          P(w_i | s_i^k) = 1, i.e., given a specific sense of a word, the surface word type is fixed, therefore P(w, c|s) = P(w|s)*p(c|s) = P(c|s).
>
> >>>
> For example, "guy" and "man" are synonym, but one is more casual and the other is more formal. Hence, despite the same sense, different realization would cause the contexts to be more (or less) formal.
> <<<
>
> In this example, “guy” and “man” won’t be associated to the same sense vector. One sense of “guy” s_{guy}^i and one sense of “man” s_{man}^j may be similar but are two separate embeddings, which would have distinct (perhaps close) P(c|s_{guy}^i) and P(c| s_{man}^j). Ideally, P(c|s_{guy}^i) would result in more casual expression while P(c| s_{man}^j) results in formal expression. Therefore, this case won’t result in the commenter’s concern.
>
>      B. When senses are shared among words
>
> In the commenter's example: it is possible that senses are divided to s_{man_casual}, s_{man_formal} and s_others, once we observe s_{man_casual}, we know that the context should be casual rather than formal no matter it’s generated by “man” or “guy”.
>
> Surely the surface word type eventually affect its context, in the way that “guy” may have a higher probability to generate s_{man_casual} while “man” may have a higher probability to generate s_{man_formal}.
>
> Although “man” and “guy” are synonyms, they will have different distribution: P(s_i| guy) and P(s_i|man).
>
> Ideally,
> P(s_{man_casual}|guy) > P(s_{man_formal}|guy),
>  and
> P(s_{man_casual}|man) < P(s_{man_formal}|man).
>
> If they have the same distribution, it means that the usage of “guy” and “man” are identical, which contradicts the commenter’s statement “one is more casual and the other is more formal”.

---

### Meta-Review · Area_Chair1 · 2018-12-13
**word sense induction with Gumbel-Softmax**

**Confidence:** 5
**Recommendation:** Reject

**Metareview:**


Pros:

*  High quality evaluation across different benchmarks, plus human eval

*  The paper is well written (though one could quibble about the motivation for the method, see Cons)

Cons:

*  The approach is incremental, the main contribution is replacing marginalization or RL with G-S. G-S has already been studied in the context of VAEs with categorical latent variables, i.e. very similar models.

*  The main technical novelty is varying amount of added noise (i.e. downscaling Gumbel noise). In principle, the Gumbel relaxation is not needed here as exact marginalization can be done (as) effectively. Unlike the standard strategy used to make discrete r.v. tractable in complex models, samples from G-S are not used in this work to weight input to the 'decoder' (thus avoiding expensive marginalization) but to weight terms corresponding to reconstruction from individual latent states (in constract, e.g., to SkimRNN of Seo et al (ICLR 2018)). Presumably adding noise to softmax helps to force sharpness on the posteriors (~ argmax in previous work) and stochasticity may also help exploration.

(Given the above, "to preserve differentiability and circumvent the difficulties in training with reinforcement learning, we apply the reparameterization trick with Gumbel softmax" seems slightly misleading)


*  With contextualized embeddings, which are sense-disambiguated given the context, learning discrete senses (which are anyway only coarse approximations of reality) is less practically important

Two reviewers are somewhat lukewarm (weak accept) about the paper (limited novelty), whereas one reviewer is considerably more positive. I do not believe that the reviews diverge in any factual information though.